# BACHVID: TRAINING-FREE VIDEO GENERATION WITH CONSISTENT BACKGROUND AND CHARACTER

A modern military command center.
A distinguished general, wearing a highly decorated uniform with gray hair.
(1) Stand. (2) Speak into a headset. (3) Discuss with officers.

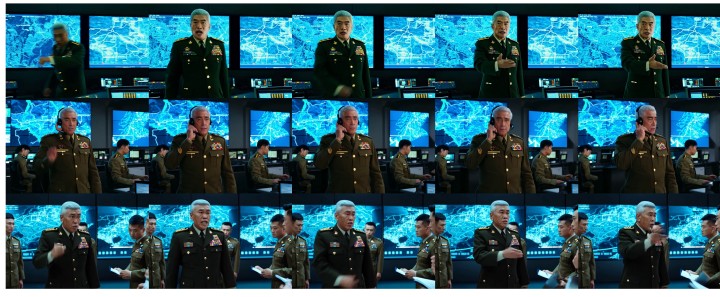

Vanilla

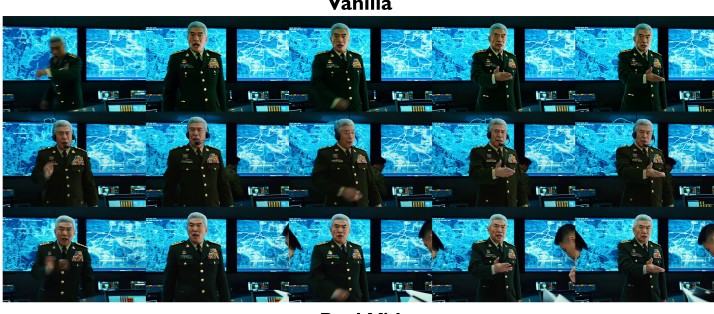

BachVid

Figure 1: We present **BachVid**, the first training-free method for **Vid**eo generation with consistent **Ba**ckground and **Ch**aracter. The three prompts share the same background (blue) and character (green) description, while the actions (red) vary. The generated videos enable consistent background and character, facilitating downstream applications such as visual storytelling.

## ABSTRACT

Diffusion Transformers (DiTs) have recently driven significant progress in text-to-video (T2V) generation. However, generating multiple videos with consistent characters and backgrounds remains a significant challenge. Existing methods typically rely on reference images or extensive training, and often only address character consistency, leaving background consistency to image-to-video models. We introduce BachVid, the first training-free method that achieves consistent video generation without needing any reference images. Our approach is based on a systematic analysis of DiT's attention mechanism and intermediate features, revealing its ability to extract foreground masks and identify matching points during the denoising process. Our method consolidates this finding by first generating an **identity video** and caching the intermediate variables, and then inject these cached variables into corresponding positions in generated videos, ensuring both foreground and background consistency across multiple videos. Experimental results demonstrate that BachVid achieves robust consistency in generated videos without requiring additional training, offering a novel and efficient solution for consistent video generation without relying on reference images or additional training.

# 1 INTRODUCTION

Text-to-video (T2V) diffusion models (Zheng et al., 2024; Yang et al., 2024; Kong et al., 2024; HaCohen et al., 2024; Wan et al., 2025) driven by Diffusion Transformers (DiTs) (Esser et al., 2024; Peebles & Xie, 2023) have made remarkable progress in recent years. While these models can ensure subject consistency within a single video, maintaining consistency across multiple videos remains challenging, particularly in applications such as storytelling and long-form video generation.

To address this issue, existing studies have proposed a variety of effective methods, which can be broadly categorized into training-based and training-free approaches. Training-based methods, such as ConsisID (Wang et al., 2025), yield strong performance but require substantial computational resources and long training time. In contrast, training-free methods avoid such costly overhead. For instance, TPIGE (Gao et al., 2025) introduces the first training-free IPT2V (Identity-Preserving Text-to-Video) framework, which eliminates both per-identity tuning during inference and additional training costs, while still retaining state-of-the-art performance. Nevertheless, these methods generally require additional reference images as input and primarily focus on preserving character identity across different backgrounds, leaving the issue of background consistency unresolved.

Meanwhile, in the image generation domain, CharaConsist (Wang et al., 2025) has demonstrated dual consistency in both background and character through a training-free method. Its core idea is to generate an identity image and cache all the key-value pairs from the DiT, which are then injected when generating new images. Yet, directly extending it to video generation still faces two challenges. First, unlike image diffusion models where the internal mechanisms of DiTs have been extensively explored (Wang et al., 2025; Avrahami et al., 2025), the inner workings of video diffusion models remain underexplored. Research on image DiTs has primarily focused on their ability to control spatial features, whereas video DiTs must additionally handle the more complex temporal dimension. To date, only DiffTrack (Wang et al., 2025) has revealed that the query-key similarity within video DiTs implicitly encodes temporal correspondences across frames, offering an important insight. Second, video latent representations are typically much larger than those in images. Blindly storing all key-value pairs of the DiT would result in severe out-of-memory (OOM) issues.

To address these challenges, we present **BachVid**, a training-free and reference-image-free method designed to ensure **Ba**ckground and **Ch**aracter consistency across multiple video generations. *First*, we extract the foreground mask from the attention weights between the text prompt and the video from specific layers at certain timesteps. *Second*, we identify the matching points between two video generation processes from the attention outputs from specific layers at certain timesteps. *Third*, we determine a subset of vital layers for key-value injection to save memory while maintaining consistency between two video generation processes. We provide a detailed analysis of the open-source video DiT models (Yang et al., 2024). The experiments demonstrate that BachVid generates videos with consistent background and character across DiT-based video generation models. Through the analysis, we uncover several key findings: 1) A few specific layers play a dominant role in extracting foreground mask, key-value injection, and identifying matching points between two generation processes; 2) Mask extraction and matching point identification strength during the early timesteps, but degrade towards the end, during the denoising process.

In summary, our contributions are as follows:

- We propose the first training-free method for text-to-video generation with consistent backgrounds and characters.
- We establish a systematic analysis for DiT-based video generation models to automatically extract foreground mask of the generated video, identify matching points between two generated videos, and determine the vital layers for key-value injection.

# 2 RELATED WORK

## 2.1 DIFFUSION-BASED T2V GENERATION

Early diffusion systems were predominantly built on UNet backbones that interleave convolution and self-attention, and inject textual guidance via cross-attention from text encoders (e.g., CLIP),

thereby enabling controllable synthesis. Diffusion Transformers (DiTs) (Peebles & Xie, 2023) replace the UNet with a Transformer (Vaswani et al., 2017) and exhibit strong scaling with data and model size. Following this trajectory, a series of DiT-based methods have achieved state-of-the-art image and video quality for both T2I (Blattmann et al., 2023; Esser et al., 2024; Labs et al., 2025) and text-to-video (T2V) generation (Yang et al., 2024; Wan et al., 2025). In contrast to UNet-based pipelines that explicitly decouple self- and cross-attention, DiTs unify attention over a joint sequence, which makes many UNet-oriented editing or control strategies—often designed to manipulate specific encoder/decoder stages—non-trivial to port to the Transformer setting.

Understanding *where* and *when* to inject conditioning has therefore become a key question. A body of work has probed internal representations of UNet-based image diffusion models (Jin et al., 2025; Meng et al., 2024; Zhang et al., 2023; Tang et al., 2023; Nam et al., 2024), largely focusing on image-space correspondences or two-frame relationships. Moving to video, DiffTrack (Nam et al., 2025) establishes temporal correspondences during denoising and reports a characteristic evolution: temporal matching strengthens through the mid timesteps (as motion and layout consolidate) but can diminish near the end when attention re-focuses on refining appearance details, while early steps rely more heavily on text and intra-frame cues to set global semantics and structure.

For DiT-based generators, StableFlow (Avrahami et al., 2025) further observes that "vital" layers for effective control are distributed across the Transformer stack rather than localized, complicating the choice of layers and timesteps for intervention. Taken together, these findings suggest that T2V models—especially Transformer-based ones—benefit from conditioning schemes that account for (i) the unified multimodal attention in DiTs, (ii) the temporal evolution of correspondences across denoising, and (iii) the dispersed importance of layers over depth and time.

## 2.2 TRAINING-FREE CONSISTENT GENERATION

Consistent generation has been extensively studied in both image (Liu et al., 2025; Zhou et al., 2024; Tewel et al., 2024; Wang et al., 2025; Mao et al., 2025; Li et al., 2024) and video domains (Wu et al., 2024; Wei et al., 2024), with most approaches relying on reference images or videos as input. MotionBooth (Wu et al., 2024) and DreamVideo (Wei et al., 2024) address per-identity consistency by fine-tuning models or incorporating additional modules. To overcome the limitations of per-ID fine-tuning, subsequent works (Mao et al., 2025; Li et al., 2024; Yuan et al., 2025; Jiang et al., 2025) propose tuning-free strategies. However, these methods typically require large-scale datasets and computationally expensive training. As a result, training-free approaches have gained increasing attention. For example, TPIGE (Gao et al., 2025) enables identity-preserving video generation without training, but its scope is limited to facial identity. In contrast, our method is also training-free but achieves both background and character consistency across generated videos.

## 3 METHODOLOGY

In this section, we first introduce some preliminaries on video generation models, outlining the fundamental architecture and mechanisms that underpin consistency across frames. Motivated by CharaConsist (Wang et al., 2025), our core idea for ensuring consistency is to establish a stable **identity video generation** and guide subsequent **frame video generation** through information reuse. The identity video provides stable representations, whose intermediate variables (keys, values, and attention outputs) are cached and later injected into the frame generation process, thereby enforcing consistency. Building on this idea, our method addresses three key components: (1) extraction of foreground and background masks, (2) identification of matching points between the identity and frame videos, and (3) injection of the identity video's keys and values into the corresponding positions of the frame videos. These components are elaborated in the following subsections.

### 3.1 PRELIMINARIES

**Video Diffusion Models.** Video diffusion models (Yang et al., 2024; Wan et al., 2025; Kong et al., 2024; Zheng et al., 2024) generate videos from the input text prompt through iterative denoising. They are typically composed of a 3D Variational Autoencoder (VAE), a text encoder, and a denoising network $f_\theta$. The 3D VAE compresses a video $Z_{\text{video}} \in \mathbb{R}^{T' \times H' \times W' \times 3}$ into a latent representation $z_{\text{video}} \in \mathbb{R}^{T \times H \times W \times C}$, where $T < T', H < H', W < W'$, for computational efficiency. The

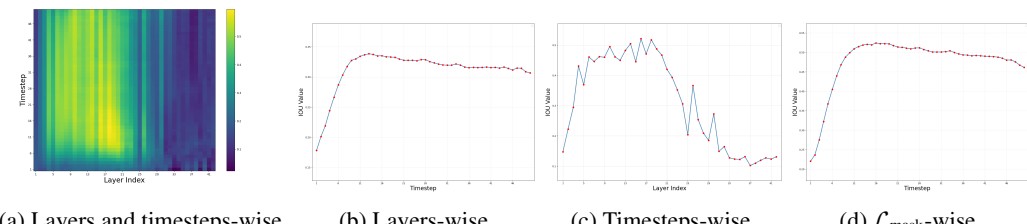

(a) Layers and timesteps-wise.    (b) Layers-wise.    (c) Timesteps-wise.    (d) $\mathcal{L}_{\text{mask}}$-wise.

Figure 2: Average IoU Evaluation of foreground mask extraction at different timesteps and layers.

text encoder maps the input prompt into an embedding $z_{\text{text}}$ with sequence length $L_{\text{text}}$. Given a predefined noise schedule, Gaussian noise is progressively added to $z_{\text{video},t}$, yielding noisy latents $z_{\text{video},t+1}$. At each step, the denoising network $f_\theta(z_{\text{video}}, t, z_{text,t})$ is trained to predict and remove the injected noise. Starting from Gaussian noise $z_{\text{video},T}$, the model iteratively denoises until reaching $z_{\text{video},0}$, which is then decoded by the 3D VAE to produce the final video $Z'_{\text{video}}$.

**Video Diffusion Transformers.** Building on the success of Sora (Liu et al., 2024b), Diffusion Transformers (DiTs) (Blattmann et al., 2023; Peebles & Xie, 2023) has become the standard backbone for video generation (Zheng et al., 2024; HaCohen et al., 2024; Yang et al., 2024; Kong et al., 2024; Wan et al., 2025). DiTs employ full 3D attention across space and time, enabling rich interactions between visual and textual representations. We illustrate this using CogVideoX (Yang et al., 2024). At each timestep $t$ and layer $l$, self-attention is applied to the concatenated sequence $z = z_{\text{video}} \oplus z_{\text{text}}$. This sequence is projected into queries $Q^{t,l}$, keys $K^{t,l}$, and values $V^{t,l}$, each in $\mathbb{R}^{(THW+L_{\text{text}}) \times C}$. The attention output is computed as:

$$O^{t,l} = W^{t,l}V^{t,l}, \quad W^{t,l} = \text{Softmax}(Q^{t,l}(K^{t,l})^T/\sqrt{C}), \tag{1}$$

where $W^{t,l} \in \mathbb{R}^{(THW+L_{\text{text}}) \times (THW+L_{\text{text}})}$ denotes the attention weights, capturing interactions between video and text latents. For instance, the video-to-text cross-attention weights $W_{\text{v2t}}^{t,l} \in \mathbb{R}^{THW \times L_{\text{text}}}$ characterize the correspondence between video tokens and text tokens.

### 3.2 Foreground Mask Extraction

To ensure consistent backgrounds and characters, it is necessary to localize the foreground (character) and background regions. Foreground extraction from complete videos is straightforward with off-the-shelf methods. However, in our setting, we must identify the foreground *during the denoising process*, where only latent features are available.

The video-to-text attention weights $W_{\text{v2t}}^{t,l}$ capture the relationship between each pixel and each text token. We structure the prompts in the form "[Background],[Character],[Action]", from which the token lengths $L_{\text{bg}}, L_{\text{fg}}$ and $L_{\text{act}}$ can be obtained, such that $L_{\text{text}} = L_{\text{bg}} + L_{\text{fg}} + L_{\text{act}} + L_{\text{pad}}$. The corresponding attention segments are then extracted as:

$$W_{\text{bg}}^{t,l} = W_{v2t}^{t,l}[:, : L_{\text{bg}}] \in \mathbb{R}^{THW \times L_{\text{bg}}}, \quad W_{\text{fg}}^{t,l} = W_{v2t}^{t,l}[:, L_{\text{bg}} : L_{\text{bg}} + L_{\text{fg}}] \in \mathbb{R}^{THW \times L_{\text{fg}}}. \tag{2}$$

A foreground mask is obtained by comparing the averaged attention weights:

$$\mathcal{M}^{t,l} = \left( \frac{1}{L_{\text{bg}}} \sum_i W_{\text{bg}}^{t,l}[:, i] \right) \le \left( \frac{1}{L_{\text{fg}}} \sum_i W_{\text{fg}}^{t,l}[:, i] \right), \tag{3}$$

where $\mathcal{M}^{t,l} \in \mathbb{R}^{THW}$ indicates whether each pixel is classified as foreground or background.

For every layer $l$ and timestep $t$, we evaluate $\mathcal{M}^{l,t}$ against the ground-truth mask $M_{gt}$ using IoU (Fig. 2a) and the results show that not all layers or timesteps contribute equally: some yield strong masks, others degrade performance. Fig. 2b shows IoU averaged over layers, and we observe that **mask quality improves during denoising but degrades at the final steps.**, as early noisy latents hinder extraction, while late steps mainly refine details rather than structure. Fig. 2c shows IoU averaged over timesteps, and we find that **early layers facilitate accurate mask extraction, while**

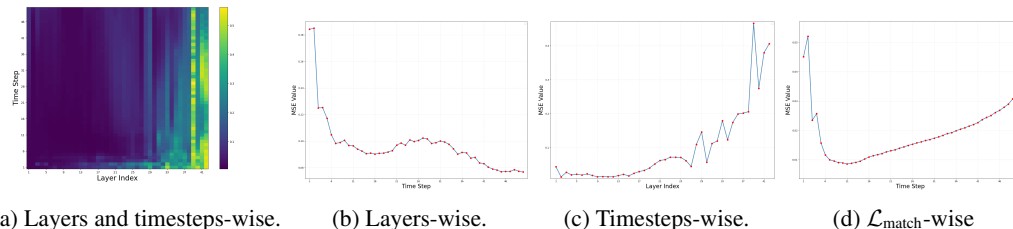

(a) Layers and timesteps-wise.  (b) Layers-wise.  (c) Timesteps-wise.  (d) $\mathcal{L}_{\text{match}}$-wise

Figure 3: MSE Evaluation of matching point identification at different timesteps and layers.

**later layers degrade it**, since encoder-like early layers capture high-level semantics, while decoder-like later layers focus on noise prediction.

Based on these observations, we select the top-15 layers (indices 6-20), denoted $\mathcal{L}_{\text{mask}}$. Evaluating on this subset (Fig. 2d) shows consistent trends. We set $\tau_{\text{mask}} = 5$ and the final robust foreground mask is thus:

$$\mathcal{M} = \sum_{l \in \mathcal{L}_{\text{mask}}} \left( \frac{1}{L_{\text{bg}}} \sum_i W_{\text{bg}}^{\tau_{\text{mask}},l}[:,i] \right) \leq \sum_{l \in \mathcal{L}_{\text{mask}}} \left( \frac{1}{L_{\text{fg}}} \sum_i W_{\text{fg}}^{\tau_{\text{mask}},l}[:,i] \right). \tag{4}$$

### 3.3 MATCHING POINT IDENTIFICATION

Consistent characters across videos require identifying correspondences between the identity and frame videos. While off-the-shelf feature matchers can be used on fully rendered videos, our task requires identifying correspondences *during denoising*, when only latents are available.

Prior work (Tang et al., 2023; Wang et al., 2025) demonstrated that semantic correspondences can be extracted by comparing intermediate diffusion features at specific timesteps. Following this, we use attention outputs $O^{t,l} \in \mathbb{R}^{THW \times C}$ for point matching.

Given $O_{id}^{t,l}$ (identity video) and $O_{frm}^{t,l}$ (frame video), we compute the frame-wise cosine similarity between frame and identity pixels:

$$S^{t,l} = \hat{O}_{frm}^{t,l} (\hat{O}_{id}^{t,l})^T \in \mathbb{R}^{T \times HW \times HW}, \tag{5}$$

where $\hat{O}_{frm}^{t,l}, \hat{O}_{id}^{t,l} \in \mathbb{R}^{T \times HW \times C}$ are normalized and reshaped from $O_{frm}^{t,l}, O_{id}^{t,l}$. The matching point for pixel $(i,j)$ in the frame video is:

$$\hat{\text{map}}(S^{t,l}, i, j) = \text{argmax}(S^{t,l}[i,j,:]). \tag{6}$$

We evaluate each $(t, l)$ against ground-truth correspondences using MSE (Fig. 3a), and the results show that not all layers or timesteps are useful. Fig. 3b shows MSE averaged over layers, and we observe that point matching improves during denoising but fluctuates due to noisy early latents. Fig. 3c shows MSEs averaged over timesteps and we observe that **early layers provide robust features, while later layers degrade performance**, mirroring mask extraction trends.

Based on these observations, we select the bottom-15 layers (indices 2-16), denoted $\mathcal{L}_{\text{match}}$, and further evaluate on these layers (See Fig. 3d). The results reflect a different trend from Fig. 3b, as the other layers badly disrupt the point matching. The observation is similar to Fig. 2d that **point matching quality improves during denoising but degrades at the final steps**.

We also set $\tau_{\text{match}} = 10$ and the final robust similarity matrix is:

$$S_{\text{match}} = \sum_{l \in \mathcal{L}_{\text{match}}} O_{frm}^{\tau_{\text{match}},l} (O_{id}^{\tau_{\text{match}},l})^T \in \mathbb{R}^{THW \times THW}. \tag{7}$$

For the $j$-th pixel in the frame video, the matching point is identified by:

$$\text{map}(S_{\text{match}}, j) = \text{argmax}(S_{\text{match}}[j,:]), \tag{8}$$

where $\text{map}(S_{\text{match}}, j) = k$ means that $j$-th pixel in the frame video and $k$-th pixel in the identity video are matching points.

Figure 4: **Overview of BachVid.** An identity video is first generated to cache key intermediate variables. For every frame video, these cached key-values are injected into matched points (Sec. 3.23.3 to ensure both foreground and background consistency, using only vital layers (Sec. 3.4).

## 3.4 VITAL LAYERS DETERMINATION

Having obtained masks and correspondences, we must decide which layers to apply key-value injection. Naively storing all key-value across timesteps and layers is infeasible for DiT-based video models (e.g., CogVideoX), as their large depth and latent dimension cause memory issues.

To reduce memory, we identify *vital layers*. Following StableFlow (Avrahami et al., 2025), which measured DINOv2 feature similarity for the stable editing task, we instead evaluate the aesthetic score of generated videos when skipping each layer. For video $Z^{-l}$ generated by skipping layer $l$, the score is:

$$\text{Score}_{\text{aes}} = \frac{1}{T'} \sum_{i=1}^{T'} \mathcal{A}(Z^{-l}[i]), \tag{9}$$

where $\mathcal{A}$ is the pre-trained aesthetic predictor.

As shown in Fig. 9, skipping layer $l \in \mathcal{L}_{\text{aes}} = \{1, 2, 12, 13, 14, 15, 16, 18, 20, 21, 22, 24, 30, 35, 42\}$ significantly reduces scores. We therefore set $\mathcal{L}_{\text{kv}} = \mathcal{L}_{\text{aes}}$ as the determined vital layers.

## 3.5 CONSISTENT BACKGROUND AND CHARACTER VIDEO GENERATION

Fig. 4 illustrates the full pipeline. During identity video generation, we extract the foreground mask $\mathcal{M}_{id}$, the attention outputs $\{O_{id}^{\tau_{\text{match}}, l}\}_{l \in \mathcal{L}_{\text{match}}}$, and store the key-values $\{K_{\text{id}}^{t,l}, V_{\text{id}}^{t,l}\}_{t \in \mathcal{T}}^{l \in \mathcal{L}_{kv}}$ across timesteps $\mathcal{T}$. During frame video generation, for each timestep $t$, we compute $\mathcal{M}_{frm}$ from $\mathcal{L}_{\text{mask}}$ and $\{O_{frm}^{\tau_{\text{match}}, l}\}_{l \in \mathcal{L}_{\text{match}}}$. We then calculate the mapping $\text{map}(S_{\text{match}}, \cdot)$ using Eq. 7- 8 and obtain the indices to the foreground and background of the frame video:

$$I_{\text{frm,fg}} = \text{nonzero}(\mathcal{M}_{\text{frm}}), \quad I_{\text{frm,bg}} = \text{iszero}(\mathcal{M}_{\text{id}} \cup \mathcal{M}_{\text{frm}}) \tag{10}$$

where non-zero$(\cdot)$ and iszero$(\cdot)$ refer to get the indices of non-zero and zero elements, respectively. Then we identify their matching points' indices in the identity video:

$$I_{\text{id,fg}} = \text{map}(S_{\text{match}}, I_{\text{frm,fg}}), \quad I_{\text{id,bg}} = I_{\text{frm,bg}}. \tag{11}$$

Based on the indices, we extract the keys and values $K_{\text{id,fg}}, V_{\text{id,fg}}, K_{\text{id,bg}}, V_{\text{id,bg}}$ (eliminate $t, i$ for simplication) from the identity image and re-encode the keys with RoPE (Su et al., 2024):

$$K'_{\text{id,fg}} = \text{RoPE}(K_{\text{id,fg}}, I_{\text{frm,fg}}), \quad K'_{\text{id,bg}} = \text{RoPE}(K_{\text{id,bg}}, I_{\text{frm,bg}}), \tag{12}$$

We then inject the key-values of the identity video into the frame video to get the fused key-values:

$$K^*_{\text{frm}} = K'_{\text{frm}} \oplus K'_{\text{id,fg}} \oplus K'_{\text{id,bg}}, \quad V^*_{\text{frm}} = V_{\text{frm}} \oplus V_{\text{id,fg}} \oplus V_{\text{id,bg}}. \tag{13}$$

Finally, the attention is computed as:

$$O^*_{\text{frm}} = W^*_{\text{frm}} V^*_{\text{frm}}, \quad W^*_{\text{frm}} = \text{Softmax}(Q_{frm}(K^*_{frm})^T / \sqrt{C}) + \log(\mathcal{M}^*), \tag{14}$$

where the attention mask $\mathcal{M}^*$ enforces that frame foreground tokens attend only to identity foreground tokens, and background to background.

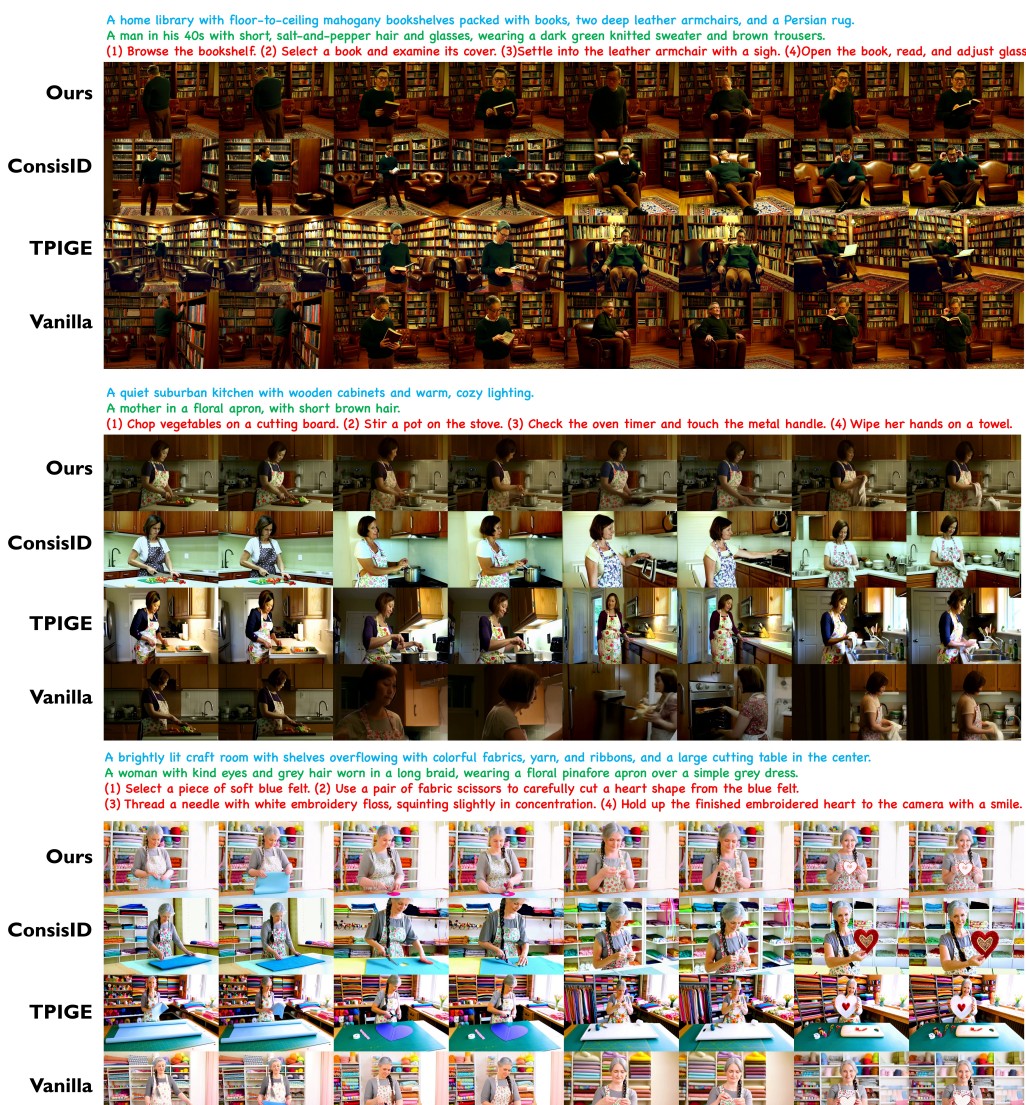

Figure 5: **Qualitative results.** We select two frames for each action for visualization.

## 4 EXPERIMENT

### 4.1 IMPLEMENTATION DETAILS

**Dataset.** While existing benchmarks lack features suited for consistent generation for characters and background, we used DeepSeek (Liu et al., 2024a) to generate a series of T2V prompts. Specifically, we instructed DeepSeek to create multiple groups of prompts in the format: "[Background], [Character], [Action].", with contents varying across groups. In each group, both "[Background]" and "[Character]" remain consistent, while "[Action]" varies. In total, we generate 40 groups with 5 prompts each, which is sufficient as the validation dataset's scale is similar to (Wang et al., 2025).

**Metric.** We follow the metrics of CharaConsist (Wang et al., 2025) and TPIGE (Gao et al., 2025). We evaluate the method from four perspectives: text alignment, background consistency, identity consistency, and video quality. For **text alignment**, we use the CLIP score (CLIP-T) to evaluate the text-video similarity. For **background consistency**, we use the PSNR and CLIP scores (PSNR-BG and CLIP-BG) of pairwise background region videos segmented by SAM2 (Ren et al., 2024; Ravi et al., 2024). For **identity consistency**, we use the CLIP scores (CLIP-FG) of pairwise foreground

Table 1: **Quantitative results with Baselines.** All metrics are higher-is-better, except for TCG which is lower-is-better.

| Method | CLIP-T (%) | PSNR-BG (dB) | CLIP-BG (%) | CLIP-FG (%) | Face-Arc (%) | MS (%) | IQ (%) | TCG |
|---|---|---|---|---|---|---|---|---|
| CogVideoX | 25.99 | 28.95 | 94.85 | 92.86 | 22.62 | 98.57 | 64.67 | 113/3/33 |
| ConsisID | 26.03 | 28.68 | 95.02 | 94.13 | 39.54 | 98.10 | 69.36 | 115/3/37 |
| TPIGE | 26.05 | 29.15 | 95.43 | 93.59 | **41.70** | **98.91** | **70.67** | 593/77/76 |
| Ours | **26.14** | **31.96** | **97.31** | **95.06** | 40.25 | 98.69 | 64.72 | 214/175/40 |

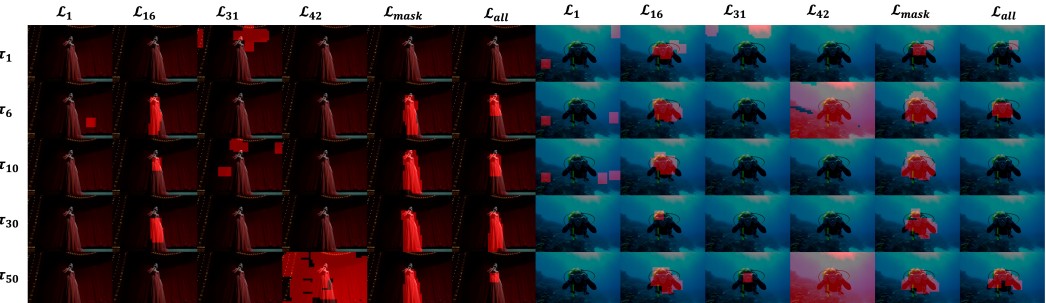

Figure 6: Mask extraction over different layers and timesteps.

region videos and compute facial embedding similarity (Face-Arc) between each generated frame and the reference image using RetinaFace and ArcFace feature spaces. For **video quality**, we use Motion Smoothness and Imaging Quality from VBench (Huang et al., 2024). For **efficiency**, we record the inference Time used (seconds)/CPU memory (GB)/GPU memory (GB), denoted as **TCG**.

## 4.2 COMPARISON WITH SOTA

As no existing methods enable generating video with a consistent background and character, we compare with methods that focus on identity-preserving. We compare with the following baselines: 1) ConsisID (Yuan et al., 2025), based on CogVideoX-5B (Yang et al., 2024), proposed a hierarchical training strategy with frequency-decomposed facial features to achieve tuning-free IPT2V. 2) TPIGE (Gao et al., 2025), based on VACE (Jiang et al., 2025), proposed to enhance face-aware prompts and prompt-aware reference images, and ID-aware spatiotemporal guidance to achieve training-free IPT2V. Because both baselines require a reference face image, we crop the character headshots using RetinaFace (Deng et al., 2020) from the identity video as their input.

Qualitative and quantitative results are shown in Fig. 5 and Table 1. Compared with SOTA approaches, our method achieves better performance on text alignment and background consistency across all methods. For identity consistency, we can obtain Face-Arc scores comparable to reference-based baselines. While ConsisID and TPIGE explicitly inject identity priors via a face image, text-to-video models often fail to generate consistently clear facial details (side views, occlusions, accessories, etc.), limiting the effectiveness of such priors. Our method injects priors implicitly at the feature level and thus achieves comparable results. Besides, we perform the best CLIP-FG, which shows that our method produces videos with more consistent overall appearance (face, clothing, style). Furthermore, our method will not degrade the video quality compared to the vanilla CogVideoX.

## 4.3 ABLATION STUDY

**How do the layer and timestep affect the foreground mask extraction?** Fig. 6 presents a visualization of mask extraction across different layers and timesteps. We observe that aggregating features from all layers often leads to suboptimal results, as certain layers (e.g., 1, 31, 42) introduce noise that interferes with accurate mask extraction. In contrast, aggregating features only from the selected layers $\mathcal{L}_{mask}$ yields a clean and consistent mask. In addition, mask extraction tends to be unreliable at the early stages of denoising, while performing it too late can also degrade the results.

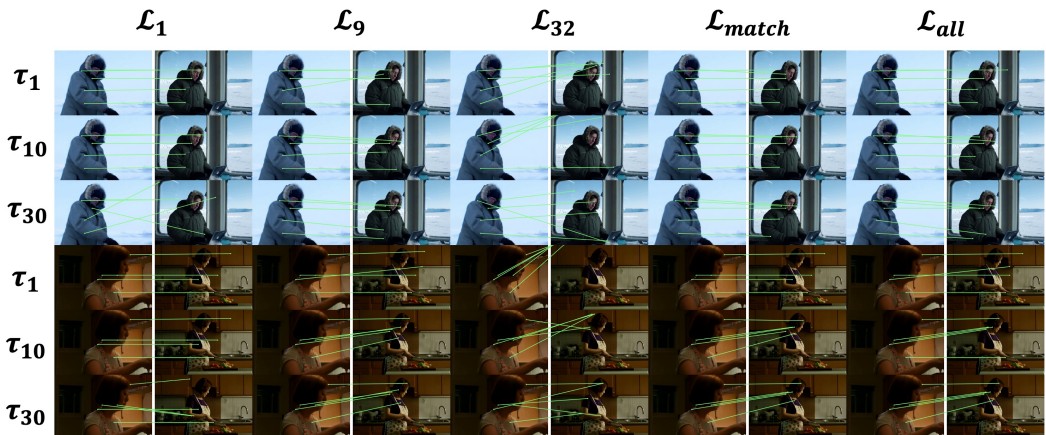

Figure 7: Matching point identification over different layers and timesteps.

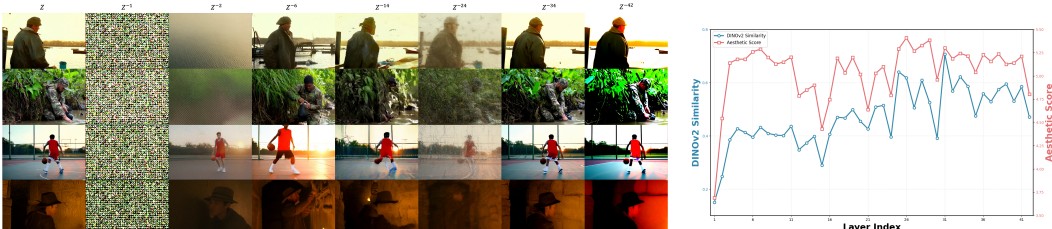

Figure 8: Visualization of $Z^{-l}$.

Figure 9: Analysis on vital layers.

**How do the layer and timestep affect the matching point identification?** Fig. 7 presents a visualization of point matching identification across different layers and timesteps. We find that aggregating features from all layers often produces suboptimal results, as certain layers (e.g., 1, 32) introduce interference that disrupts point matching. In contrast, using features only from the selected layer $\mathcal{L}_{\text{match}}$ yields more robust and reliable matching points. Moreover, point matching is also sensitive to timing: performing it too early or too late in the denoising process is unstable.

**How does the layer affect the KV Injection?** We also conduct an ablation study on the metrics for selecting vital layers. To preserve semantic content, StableFlow (Avrahami et al., 2025) identifies vital layers for image editing using DINOv2 similarity:

$$\text{Score}_{\text{DINOv2}} = \frac{1}{T'} \sum_{i=1}^{T'} \text{CosSim}(\mathcal{E}(Z[i]), \mathcal{E}(Z^{-l}[i]), \tag{15}$$

where $\mathcal{E}$ is the pre-trained DINOv2 encoder. Fig. 9 presents the results and we denote $\mathcal{L}_{\text{DINOv2}}$ as the 15 vital layers. We notice that skip layer $l \in \{3, 5, 6, 8, 9, 10\}$ results in low DINOv2 similarity but keeps a high aesthetic score. This is because the generated videos still keep moderate quality, but show different diversity (see $Z^{-6}$ in Fig. 8). We infer that these layers mainly control motion diversity and should therefore be treated as *non-vital.* Table. 2 quanlitatively demonstrate that injecting KV of $\mathcal{L}_{\text{aes}}$ performs better than $\mathcal{L}_{\text{DINOv2}}$.

Table 2: **Ablation study on $\mathcal{L}_{\text{kv}}$.** All metrics are higher-is-better.

| $\mathcal{L}_{\text{kv}}$ | CLIP-T (%) | PSNR-BG (dB) | CLIP-BG (%) | CLIP-FG (%) | Face-Arc (%) | MS (%) | IQ (%) |
|---|---|---|---|---|---|---|---|
| $\mathcal{L}_{\text{DINOv2}}$ | 26.12 | 31.18 | 96.86 | 94.24 | 32.99 | 98.76 | 65.13 |
| $\mathcal{L}_{\text{aes}}$ | **26.13** | **31.69** | **97.03** | **94.24** | **36.93** | **98.77** | **65.22** |

Table 3: **Quantitative results with different DiT-based models.** All metrics are higher-is-better.

| Method | CLIP-T (%) | PSNR-BG (dB) | CLIP-BG (%) | CLIP-FG (%) | Face-Arc (%) | MS (%) | IQ (%) |
|---|---|---|---|---|---|---|---|
| CogVideoX-5B | 25.99 | 28.95 | 94.85 | 92.86 | 22.62 | 98.57 | 64.67 |
| Ours(CogVideoX-5B) | 26.13 | 31.69 | 97.03 | 94.24 | 36.93 | 98.77 | 65.22 |
| Wan2.1-T2V-14B | 26.22 | 28.76 | 95.03 | 92.08 | 18.20 | 99.08 | 70.29 |
| Ours(Wan2.1-T2V-14B) | 26.28 | 31.62 | 97.52 | 94.70 | 32.54 | 98.95 | 69.78 |

**More DiT-based video models.** To further demonstrate the effectiveness of our method, we conduct additional analyses and experiments on another DiT-based model: Wan2.1-T2V-14B (Wan et al., 2025). The architecture of Wan differs from that of CogVideoX, notably in its use of cross-attention to integrate textual prompts into video latents. The corresponding results are presented in Tab. 3, with a detailed analysis provided in Sec. B. Overall, maintaining consistency solely through textual descriptions of the background and characters is insufficient to ensure content consistency in generated videos. Our method, however, effectively achieves this.

**How do $\tau$ and the number of chosen layers affect performance?** We conduct ablation studies on $\tau_{\text{mask}}$ and $\tau_{\text{match}}$, as well as on the number of selected layers used for $\mathcal{L}_{\text{mask}}$, $\mathcal{L}_{\text{match}}$, and $\mathcal{L}_{\text{kv}}$. The results are summarized in Tab. 4. For simplicity, we set $\tau = \tau_{\text{mask}} = \tau_{\text{match}}$, and use $|\mathcal{L}|$=n to denote the selection of $n$ layers for each of $\mathcal{L}_{\text{mask}}$, $\mathcal{L}_{\text{match}}$, and $\mathcal{L}_{\text{kv}}$. SR refers to the success rate of generating all frame videos. In our setup, failure typically occurs when the extracted foreground mask is entirely False, preventing BachVid from generating valid videos. Setting $\tau = 3$ tends to produce inaccurate masks and mismatched points, leading to a lower success rate. As $\tau$ increases, inference time also rises, while larger values of $|\mathcal{L}|$ result in higher memory consumption. To achieve a better balance between output quality and computational efficiency, we ultimately select $\tau = 5$ and $|\mathcal{L}| = 15$.

Table 4: **Ablation studies on $\tau$ and $|\mathcal{L}|$.** The bolded metrics indicate the optimal values and the underlined metrics indicate the sub-optimal values, except for those at $\tau$=3.

| $\tau, |\mathcal{L}|$ | CLIP-T (%) | PSNR-BG (dB) | CLIP-BG (%) | CLIP-FG (%) | Face-Arc (%) | MS (%) | IQ (%) | TCG | SR(%) |
|---|---|---|---|---|---|---|---|---|---|
| 3, 10 | 26.13 | 31.36 | 97.31 | 95.59 | 42.42 | 98.68 | 65.44 | 172/118/35 | 94.37 |
| 3, 15 | 26.15 | 31.96 | 97.59 | 96.05 | 45.71 | 98.66 | 65.46 | 199/175/39 | 93.13 |
| 3, 20 | 26.17 | 32.22 | 97.47 | 95.91 | 46.33 | 98.64 | 65.37 | 226/232/45 | 91.87 |
| 5, 10 | 26.13 | 31.34 | 97.01 | 94.82 | 37.42 | 98.71 | 64.97 | 184/118/35 | 99.38 |
| 5, 15 | 26.14 | 31.96 | **97.31** | 95.06 | 40.25 | 98.69 | 64.72 | 214/175/40 | 99.38 |
| 5, 20 | 26.12 | **32.24** | 97.27 | **95.17** | **43.67** | 98.70 | 64.80 | 244/232/45 | 99.38 |
| 10, 10 | **26.16** | 31.06 | 96.77 | 94.41 | 35.42 | 98.78 | 65.18 | 205/118/35 | 100.00 |
| 10, 15 | 26.13 | 31.69 | 97.03 | 94.24 | 36.93 | 98.77 | 65.22 | 241/175/41 | 100.00 |
| 10, 20 | 26.13 | 31.96 | 97.04 | 94.58 | 38.50 | 98.77 | 64.97 | 278/232/45 | 100.00 |
| 15, 10 | 26.11 | 30.95 | 96.60 | 94.07 | 31.69 | **98.79** | 65.20 | 226/117/35 | 100.00 |
| 15, 15 | 26.12 | 31.59 | 96.90 | 94.15 | 33.17 | **98.79** | 65.20 | 264/175/39 | 99.38 |
| 15, 20 | 26.11 | 31.86 | 96.96 | 94.21 | 33.74 | 98.78 | 65.17 | 304/232/45 | 100.0 |

## 5 CONCLUSION

In this work, we presented BachVid, the first training-free approach for generating multiple videos with consistent characters and backgrounds, without relying on reference images or additional training. By systematically analyzing the attention mechanism and intermediate features of DiTs, we revealed their intrinsic ability to extract foreground masks and identify matching points during denoising. Building on these insights, BachVid generates an identity video, caches key intermediate variables, and reinjects them into subsequent generations to ensure both foreground and background consistency. Experiments confirm that our method achieves robust consistency while remaining efficient and training-free.

**Limitation and future work.** BachVid does not support reference identity or background images as inputs. A promising direction is to integrate our findings on mask extraction, point matching, and vital layer selection into reference image–based methods, which could further enhance their training efficiency.

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

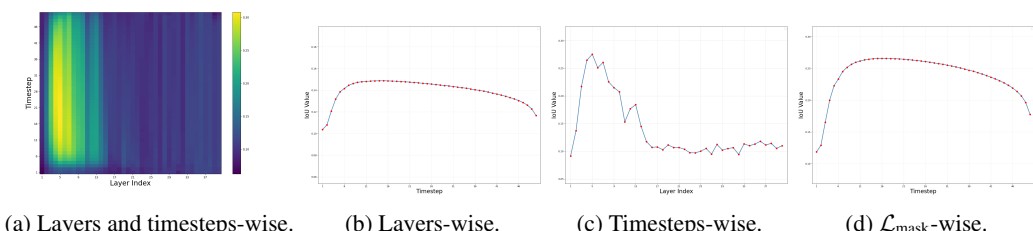

(a) Layers and timesteps-wise. (b) Layers-wise. (c) Timesteps-wise. (d) $\mathcal{L}_{\text{mask}}$-wise.

Figure 10: Average IoU Evaluation of foreground mask extraction at different timesteps and layers.

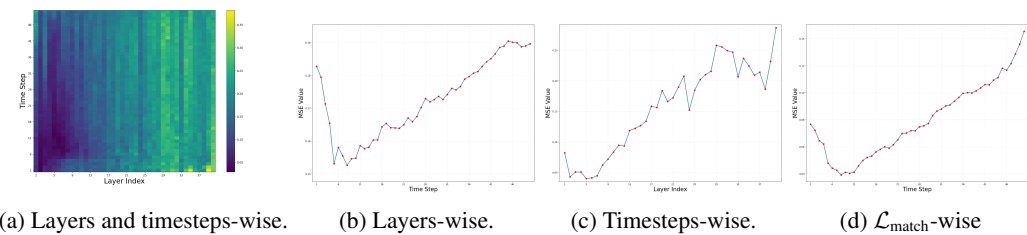

(a) Layers and timesteps-wise. (b) Layers-wise. (c) Timesteps-wise. (d) $\mathcal{L}_{\text{match}}$-wise

Figure 11: MSE Evaluation of matching point identification at different timesteps and layers.

## A USAGE OF LLM

We use ChatGPT to polish our writing, and use DeepSeek for text-to-video prompt generation, which is used as evaluation dataset.

## B RESULTS ON WAN2.1

To further prove the effectiveness of our method, we conduct analysis and experiments on another DiT-based Model: Wan2.1-T2V-14B. The architectural of Wan is different from CogVideoX. Wan uses cross-attention to integrate text prompt to video latents.

### B.1 FOREGROUND MASK EXTRACTION

For foreground mask extraction, we find that using whole sentences of [Background] and [Character] results in bad performance, because high attention weights tends to occur in words like "the", "with" and ".", etc.

Therefore, we use LLM to identify the noun of the sentences to detail the scope, the results show similar trends with CogVideoX (See Fig 10a).

We select the top-8 layers (indices 3-10), whose IoU values are above the 70% of the maximum IoU. We set $\tau_{\text{mask}} = 9$, where IoU reaches >95% of its maximum.

### B.2 MATHCING POINT IDENTIFICATION

For matching point identification, the results are shown in Fig. 11a. Note that the results in Fig 11c are evaluated on timesteps (5-15) because we find that the other timesteps has bad results in Fig 11b.

Based on these observations, we select the bottom-6 layers (indices 2-7), denoted $\mathcal{L}_{\text{match}}$, and further evaluate on these layers (See Fig. 11d). The results reflect a different trend from Fig. 11b, as the other layers badly disrupt the point matching. The observation is similar to Fig. 10d that **point matching quality improves during denoising but degrades at the final steps.**

We also set $\tau_{\text{match}} = 9$, where MSE is within 105% of the minimum.

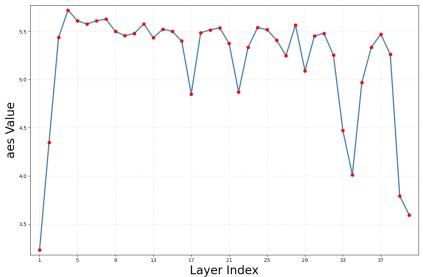

Figure 12: Aes scores when skipping layers.

### B.3 VITAL LAYERS DETERMINATION

As shown in Fig 12, skipping layer $l \in \mathcal{L}_{\text{aes}}=\{1, 2, 17, 22, 24, 29, 33, 34, 35, 39, 40\}$ significantly reduces scores.

We showcase more generated videos in Fig. 15.

## C RESULTS WITH OTHER PROMPT TEMPLATE

We further provide examples generated by prompt formed as "[Character],[Background],[Action]" in Fig. 14. The results demonstrate that our method would not be overfitting to prompt formed as "[Background],[Character],[Action]".

## D FAILURE CASE

If the identity video contains complex scene and actions, the extracted mask may not be perfect, which would result in undesirable videos. We provide two examples in Fig. 13. In the first example, the mask at the slender fishing rod in the identity video is difficult to extract completely, so the fishing rod in the generated frame video is also regarded as the "background". In the second example, there are multiple contents in the video that could be regarded as "general", making it difficult to obtain the desired mask.

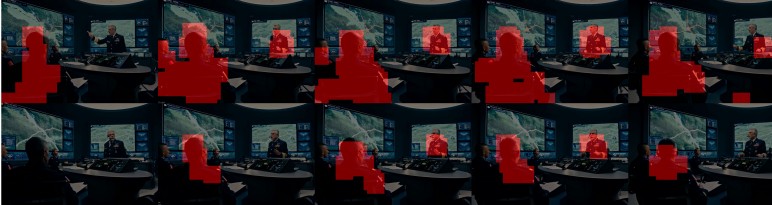

Figure 13: **Failure cases.**

A monk, dressed in traditional orange robes and with a shaved head.
A peaceful Japanese Zen garden with meticulously raked sand and blooming cherry blossom trees.
1) sits cross-legged in deep meditation under a small pagoda.
2) stands gracefully by a small tea table. He is pouring tea into a delicate, small cup.
3) stands quietly, his focused gaze on a small incense stick as he carefully lights it.

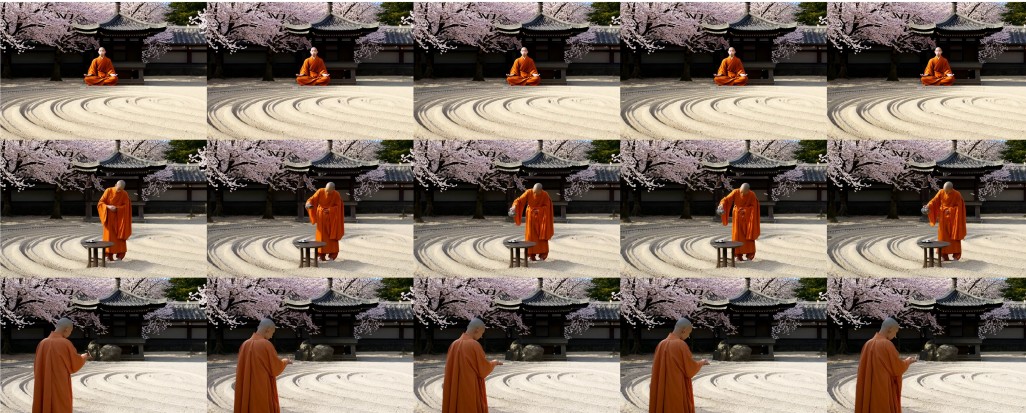

A businesswoman in a navy suit with glasses and a neat bun.
A modern office space with glass walls and glowing computer monitors, the lighting is bright and the atmosphere is professional.
1) sits at her desk, sipping coffee while reviewing documents.
2) is seated at a sleek desk, typing quickly on a laptop.
3) is standing at her desk, speaking on a phone with a serious expression.

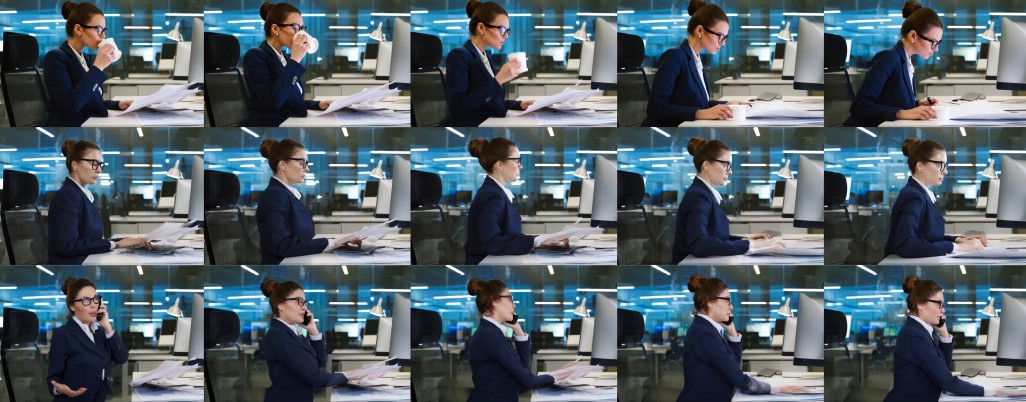

Figure 14: **Results with other prompt template.** Videos generated from prompts formed as "[Character],[Background],[Action]".

A spacious, well-organized garage workshop with a sturdy workbench, tools hanging on a pegboard, and a vintage motorcycle on a center stand.
A mechanic in her 30s with her hair tied up in a red bandana, wearing oil-stained coveralls and protective goggles.
1) selects a socket wrench from the pegboard wall.
2) peers inside the engine, using a flashlight to inspect the components.
3) smiles and gives a thumbs-up, having identified the issue.

A sunny, tranquil fishing pier extending over calm blue water, with a fishing rod holder and a cooler sitting on the wooden planks
An old fisherman with a weathered face and a bucket hat, wearing a plaid shirt and holding a fishing rod
1) places the rod in the holder and sits down on a small stool to wait.
2) watches the tip of the rod intently for any sign of a nibble.
3) reels in the line with a steady hand, a grin spreading across his face as he feels the weight of the catch.

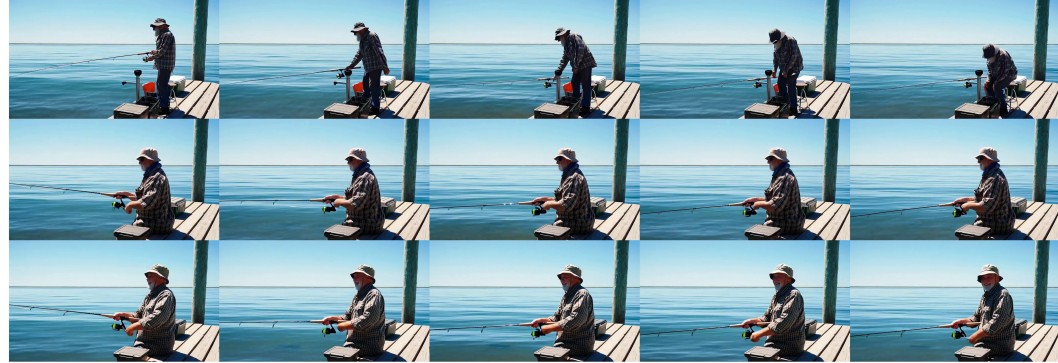

Figure 15: **More Results of BachVid (Wan2.1).**

