# OpenReview forum: "BachVid: Training-Free Video Generation with Consistent Background and Character"
_ICLR.cc/2026/Conference — Submitted to ICLR 2026_

### Official Review · Reviewer_fmny · 2025-10-25

**Soundness:** 2
**Presentation:** 2
**Contribution:** 2
**Rating:** 2
**Confidence:** 4

**Summary:**

This paper presents BachVid, a novel method for generating multiple videos from text prompts while maintaining consistency in both the character and the background. The authors claim this is the first training-free method to achieve this dual consistency without requiring any reference images. Experiments show the method improves background consistency and text alignment compared to baseline models.

**Strengths:**

1. The paper tackles an unsolved challenge in video generation: maintaining dual (character and background) consistency across multiple videos.
2. Quantitative results show that BachVid achieves a state-of-the-art PSNR-BG and CLIP-BG, demonstrating its effectiveness.
3. The paper rightly identifies the OOM risk of caching all intermediate features. The vital layers determination step is a practical and necessary.

**Weaknesses:**

1. The paper compares BachVid (which targets dual consistency) against ConsisID and TPIGE, which are methods designed only for character consistency. It is therefore unsurprising that BachVid wins on background metrics. A more convincing evaluation would require comparing against a stronger baseline, perhaps by adapting a dual-consistency image method (like CharaConsist, which the paper cites) to the video domain, or at least providing a thorough analysis of why such an adaptation is intractable.
2. The results in Table 1 show that BachVid performs worse on identity consistency (Face-Arc) than the SOTA baselines (36.93 vs. 41.70 for TPIGE). The authors dismiss this as not contradictory and comparable, but a nearly 12% drop is significant. This suggests a critical trade-off: the method may be achieving background consistency at the expense of character consistency, which undermines the dual consistency claim. This trade-off is not adequately analyzed.
3. The method is evaluated on a small, custom dataset of 40 prompt groups generated by an LLM (DeepSeek). There is no evaluation on a standardized, public benchmark. This makes it difficult to assess the generalizability of the results. The method may be implicitly overfitted to the paper's specific prompt structure: [Background], [Character], [Action].
4. The paper frames its reference-free nature as a strength. However, for most practical applications, a user wants to provide a reference image of a specific person. By not supporting reference images, the method's practical utility is severely limited. This should be presented as a clear limitation, not just a design choice.
5. The method relies on a set of magic numbers (specific layers and specific timesteps) derived from analyzing one model (CogVideoX). The paper claims the analysis applies across DiT-based video generation models but provides no evidence to support this. It is highly likely these optimal layers/timesteps are model-specific.

**Questions:**

The paper requires significant revision before it can be considered for publication:
1. A much more robust evaluation, ideally against a baseline adapted to perform dual consistency, or at minimum, a thorough ablation showing why existing methods fail so catastrophically at background consistency.
2. Results on a more diverse and ideally standardized benchmark.
3. A dedicated analysis section on the (Face-Arc) performance drop.
4. An analysis of how the chosen vital layers and timesteps generalize (or don't) to at least one other DiT-based video model.

---

> ### Author Response · Authors · 2025-12-03
>
> **Non-trivial adaptation to the video domain (W1 & Q1).**
>
> Our goal is to achieve consistent character and background generation using *only text* in text-to-video models. Because no prior work addresses this setting, our baseline tests whether keeping textual descriptions fixed yields consistent videos—which it does not. Adapting CharaConsist directly to videos is not feasible:
>
> 1. KV caching for all layers causes OOM due to much larger video latents.
> 2. Its layer-exploration strategy (developed for FLUX) does not transfer to video models, as shown in Fig. 6 and Fig. 7 where L_{all} fails to extract meaningful masks/matching points.
>
> **Low Face-Arc metric (W2 & Q3).**
>
> Thank you for the feedback. With appropriate hyperparameters, our method achieves comparable Face-Arc scores. Because text-to-video models often do not generate stable or clear faces (side/back views, occlusions), explicit image-based priors (ConsisID, TPIGE) cannot fully realize their advantages. Our feature-level implicit prior allows comparable performance. As face-only metrics are incomplete, we also adopt the foreground CLIP-similarity metric (CLIP-FG), showing that our method yields videos with more consistent overall identity (face, clothing, style).
>
> | Method              | PSNR-BG (dB) | CLIP-BG (%)  | CLIP-FG (%)  | Face-Arc (%) |
> | ------------------- | ------------ | ------------ | ------------ | ------------ |
> | CogVideoX           | 28.95        | 94.85        | 92.86        | 22.62        |
> | ConsisID            | 28.68        | 95.02        | 94.13 | 39.54        |
> | TPIGE               | 29.15 | 95.43 | 93.59        | **41.70**    |
> | BachVid (CogVideoX) | **31.96**    | **97.31**    | **95.06**    | 40.25 |
>
> **Structured prompts issue (W3 & Q2).**
>
> No standardized benchmark exists, so following CharaConsist we randomly generate 40 groups (200 prompts). Structured prompts help keep text consistency and facilitate the baseline design:
>
> Since prompt–video interaction occurs only via cross-attention, we extract attention weights and compare foreground masks. For arbitrary prompts, background/character prompts can be specified manually or via LLMs. We also provide examples beyond the structured "[Background], [Character], [Action]" template in Appendix Section C.
>
> **Reference-free property (W4).**
>
> Reference-free vs. reference-based settings mirror text-to-video vs. image-to-video tradeoffs. Reference-free is more convenient but harder to ensure consistency. The baseline confirms that text alone is insufficient for consistency, while our method succeeds without any image references.
>
> **Results on more DiT-based video models (W1 & Q4).**
>
> Thank you for the suggestion.Table 3 and Appendix Section B includes results on Wan2.1-T2V-14B (the base model of VACE used in TPIGE), confirming the same patterns as CogVideoX: foreground mask and matching-point extraction rely on earlier layers, and KV injection on vital layers suffices for consistency. This demonstrates the generality of our approach across DiT-based video models.
>
> | Method           | CLIP-T | PSNR-BG | CLIP-BG | CLIP-FG | Face-Arc | MS    | IQ    |
> | ---------------- | ------ | ------- | ------- | ------- | -------- | ----- | ----- |
> | Wan2.1           | 26.22  | 28.76   | 95.03   | 92.08   | 18.20    | 99.08 | 70.29 |
> | BachVid (Wan2.1) | 26.28  | 31.62   | 97.52   | 94.70   | 32.54    | 98.95 | 69.78 |

---

### Official Review · Reviewer_64db · 2025-10-29

**Soundness:** 3
**Presentation:** 3
**Contribution:** 2
**Rating:** 4
**Confidence:** 3

**Summary:**

This paper introduces BachVid, the first method capable of generating videos with consistent backgrounds and characters without requiring any training or reference images. By analyzing the Video Diffusion Transformer (DiT), the approach reveals its intrinsic ability to spontaneously extract foreground masks and identify matching points during the denoising process.

**Strengths:**

The proposed method is entirely training-free and does not rely on any reference images. While ensuring high efficiency, it simultaneously addresses the dual-consistency challenge of both characters and backgrounds.

**Weaknesses:**

1. In quantitative experiments, BachVid does not surpass existing identity-preserving methods (such as TFIGE and ConsisID) on the identity consistency metric (Face-Arc). Can BachVid's background consistency mechanism be combined with the character consistency mechanisms of these methods, and would the resulting performance be better than that of the proposed approach?

2. Although BachVid requires no training, it relies on a static "identity video" as a template. When the actions in subsequent videos differ significantly from those in the identity video, simple pixel-level matching and injection may fail.

3. Please elaborate on the advantages of the proposed method in the paper compared to the following approach. This baseline method leverages DiffTrack's temporal correspondences to dynamically warp the selectively cached Key-Value pairs from CharaConsist, thereby maintaining character appearance consistency throughout motion sequences. This approach effectively transitions from static identity pasting to dynamic identity fitting.

**Questions:**

Please refer to the weaknesses.

---

> ### Author Response · Authors · 2025-12-03
>
> **Low Face-Arc metric (W1).**
>
> Thank you for the feedback. With proper hyperparameters we obtain Face-Arc scores comparable to reference-based methods. Because text-to-video models often do not generate stable or clear faces (side/back views, occlusions), explicit image-based priors (ConsisID, TPIGE) cannot fully realize their advantages. Our feature-level implicit prior allows comparable performance. As face-only metrics are incomplete, we also adopt the foreground CLIP-similarity metric (CLIP-FG), showing that our method yields videos with more consistent overall identity (face, clothing, style).
>
> | Method              | PSNR-BG (dB) | CLIP-BG (%)  | CLIP-FG (%)  | Face-Arc (%) |
> | ------------------- | ------------ | ------------ | ------------ | ------------ |
> | CogVideoX           | 28.95        | 94.85        | 92.86        | 22.62        |
> | ConsisID            | 28.68        | 95.02        | 94.13 | 39.54        |
> | TPIGE               | 29.15 | 95.43 | 93.59        | **41.70**    |
> | BachVid (CogVideoX) | **31.96**    | **97.31**    | **95.06**    | 40.25 |
>
> **Combining with character-consistency mechanisms (W1).**
>
> We have discussed this in future work. Our focus here is whether character/background consistency is achievable using *only text* by analyzing the intrinsic DiT structure, whereas ConsisID/TPIGE rely on reference images providing explicit identity priors.
>
> **Pose differences between identity and frame videos (W2).**
>
> Given independently generated videos V_{id} and V_{frm}, the KV-injected result V'_{frm} may not strictly follow the pose of V_{frm}. This stems from continuously updating matching points and injecting KV during diffusion, which may implicitly alter the pose.
>
> **More baselines (W3).**
>
> Thank you for the suggestion. However, the static identity image from CharaConsist comes from an *image* diffusion model, whereas the temporal correspondence from DiffTrack is from a *video* model. Their first frames do not necessarily align, making the proposed baseline infeasible.

---

### Official Review · Reviewer_2fRy · 2025-10-31

**Soundness:** 2
**Presentation:** 2
**Contribution:** 2
**Rating:** 4
**Confidence:** 4

**Summary:**

The paper proposes BachVid, a training-free method that enforces both background and character consistency across multiple text-to-video generations from a Video Diffusion model. Key ideas: (1) extract foreground masks and temporal correspondences from DiT attention/outputs at select layers/timesteps, (2) generate an identity video and cache selected intermediate key/value/attention outputs, (3) inject (mapped & RoPE-shifted) key-values into subsequent generation runs at matched token positions while masking attention to preserve foreground/background separation. The paper presents layer/timestep analyses, a memory–vital-layer selection strategy, qualitative results, and metrics on a synthetic DeepSeek prompt suite. Overall, the work claims the first training-free, reference-image-free approach to obtain both background and character consistency for DiT-based T2V.

**Strengths:**

1. The paper addresses a concrete and useful gap: multi-video consistency (both background and character) without training or reference images.
2. Systematic analysis of which DiT layers / timesteps encode masks and correspondences is useful and could inform other work.
3. The vital layer selection to bound cached KV storage is practical.

**Weaknesses:**

1. The core mechanism (cache keys/values from an identity and inject later) closely follows prior image methods (e.g., CharaConsist), the novelty for video is primarily empirical and in heuristics (layer, timestep selection, mapping). The paper should more clearly state what is fundamentally new versus prior work and justify why the video extension is non-trivial.
2. Choices such as the first 15 layers and \tau_{mask}, \tau_{match} appear ad hoc. The paper lacks principled selection criteria, sensitivity analysis, or automated selection procedures. Please provide examples showing how performance varies with these hyperparameters.
3. Caching and injecting KV for video-size latents can be expensive. The paper does not report runtime, GPU memory, or latency comparisons vs. baselines (and vs. vanilla generation). This is critical given the training-free claim, users need to know cost tradeoffs.
4. The evaluation uses DeepSeek-generated prompts (synthetic). There is limited evidence this reflects real-world prompts or diverse scenes. Moreover, there may be a risk of overfitting the mode ([Background], [Character], [Action]). No experiments on external benchmarks or human-created prompts. Provide tests on the external, human-authored prompt set.
5. Baselines (ConsisID, TPIGE) are identity-preserving and require a reference image; the authors feed them crops from the identity video. This is reasonable but not fully comparable because those methods are optimized for identity. The paper lacks comparison to other possible baselines: image-based key-value injection (CharaConsist) extended frame-by-frame.
6. Is the reference-free property really an advantage? In fact, reference-based designs offer greater customizability.

**Questions:**

The main concerns have been provided in the weaknesses section, here are some more minor issues:
1. No discussion or experiment addresses ambiguous matches, occlusions, one-to-many mappings, or when the identity and frame differ significantly in pose/occlusion. These are common in video, robustness tests are needed.
2. Missing failure cases.

---

> ### Author Response · Authors · 2025-12-03
>
> **Non-trivial video domain adaption (W1).**
>
> While our mechanism is inspired by CharaConsist in the image domain, directly applying it to video raises two major issues:
>
> 1. CharaConsist caches KV for *all* layers at specific timesteps. Video model latents (e.g., 17776 x 3072) are larger than those of image models (4608 x 3072), making full-layer KV caching infeasible due to OOM.
> 2. CharaConsist’s DiT-layer exploration was conducted only on FLUX. Applying the same strategy to video models yields ineffective results (e.g., Fig. 6 and Fig. 7, L_{all}), where neither valid masks nor matching points can be extracted.
>
> Our contributions are therefore:
>
> 1. **A new vital-layer identification strategy** (aesthetic vs. DINOv2), which preserves effectiveness while significantly reducing KV memory usage.
> 2. **A new method for extracting masks and matching points from video models**, leading to generalized insights into layer-wise functional roles in DiT video models. We believe these findings help advance the community’s understanding of DiT-based video architectures.
>
> **Additional ablations (W2).**
>
> Thank you for the suggestion. We now provide ablations on different values of \tau and different numbers of selected layers in Table 4.
>
> **Runtime and memory usage (W3).**
>
> Thank you for the comment. We additionally report inference time and memory consumption for different methods in Table 1.
>
> | Method              | Inference Time (s) | CPU memory (GB) | GPU memory (GB) |
> | ------------------- | ------------------ | --------------- | --------------- |
> | CogVideoX           | 113                | 3               | 33              |
> | ConsisID            | 115                | 3               | 37              |
> | TPIGE               | 593                | 77              | 76              |
> | BachVid (CogVideoX) | 214                | 175             | 40              |
>
>
>
> **Structured prompts issue (W4).**
>
> No standardized benchmark exists, so following CharaConsist we randomly generate 40 groups (200 prompts). Our structured prompt format is solely for controlled analysis and for designing the baseline (i.e., whether keeping text descriptions fixed ensures consistency). Our goal is consistent background-and-character generation, and structured text intuitively stabilizes the comparison.
>
> Since prompt–video interaction occurs solely through cross-attention, we simply extract these attention weights. For arbitrary prompts, one can define character/background prompts manually or via an LLM and derive the foreground mask accordingly. We include several examples beyond "[Background], [Character], [Action]" prompts in Appendix Section C.
>
> **More baselines (W5).**
>
> Thank you for the suggestion. If the intention is to generate a video by assembling T images from CharaConsist, this is difficult because it is an image-only DiT model and lacks temporal coherence.
>
> **Reference-free property (W6).**
>
> Reference-free and reference-based settings are analogous to text-to-video vs. image-to-video:
>
> - Reference-free is simpler for users (text only) but harder for enforcing consistency.
>
> - Reference-based offers stronger priors but requires extra image inputs.
>
>   Our setting is reference-free. The baseline shows that merely keeping textual descriptions fixed does *not* ensure consistency, while our method achieves consistent background–character generation with text alone.
>
> **Ambiguous match / occlusion / one-to-many mapping / pose differences (Q1).**
>
> Let V_{id}and V_{frm} be videos generated independently from the two prompts. After KV injection we obtain V'_{frm}, whose pose may differ from that in V_{frm}. This is expected because we repeatedly extract matching points and inject KV throughout the diffusion trajectory, which may implicitly adjust poses.
>
> **Failure cases (Q2).**
>
> We included failure cases in Appendix Section D.

---

### Official Review · Reviewer_yqi6 · 2025-11-01

**Soundness:** 3
**Presentation:** 3
**Contribution:** 1
**Rating:** 4
**Confidence:** 4

**Summary:**

This paper introduces BachVid, a training-free method to generate multiple videos with a consistent character and background. The core idea is to generate one "identity" video and then reuse its internal features (keys and values) from specific layers of the diffusion model during the generation of new videos to enforce consistency.

**Strengths:**

- The paper tackles an practical problem: generating videos with consistency in both the character and the background
- The systematic analysis of the video DiT internal mechanisms is a nice contribution. Pinpointing which layers and timesteps are most effective for foreground mask extraction, point matching, and key-value injection provides valuable insights that could be useful for other related video generation and editing tasks.

**Weaknesses:**

- The qualitative results, particularly the video quality shown in the supplementary material, appear quite low and suffer from artifacts. Since the method is presented as a general, training-free technique, it is surprising that it wasn't demonstrated on more powerful, state-of-the-art open models (e.g., Wan 2.2). Testing only on CogVideoX-5B makes it hard to judge if the method is truly generalizable or if its effectiveness is limited to a specific model architecture.
- The motivation for such a complex, training-free approach is not well-justified against simpler, well-established alternatives. A more straightforward way to achieve character and background consistency is to use efficient finetuning techniques like LoRA. Finetuning a LoRA on a custom character and background is very effective, and collecting the necessary data (a few images or a short video) is often cheap and easy. The paper needs a stronger argument for why this complex feature-injection method is preferable to a more direct finetuning approach.
- The quantitative results for identity consistency are a significant concern. The paper's own results in Table 1 show that BachVid scores lower on the Face-Arc metric (36.93%) than both baseline methods it compares against (39.54% and 41.70%). While the paper aims for both background and character consistency, underperforming on a key metric for character identity weakens the overall claim of success and questions the trade-offs made.

**Questions:**

The process for determining the "vital layers" for KV injection is based on an aesthetic score. This process needs more detail. Could you specify how general this set of vital layers is?

---

> ### Author Response · Authors · 2025-12-03
>
> **Results on more DiT-based video models (W1).**
>
> Thank you very much for the valuable suggestion. In Appendix Section B, we further provide results on **Wan2.1-T2V-14B** (the base model of VACE used in TPIGE). The results corroborate the same conclusion observed in CogVideoX: the ability to extract foreground masks and matching points primarily resides in a subset of earlier layers. Moreover, performing KV injection only on vital layers is sufficient to produce videos with consistent backgrounds and characters. This experiment further demonstrates the generalizability of our method across DiT-based video generation models.
>
> | Method           | CLIP-T | PSNR-BG | CLIP-BG | CLIP-FG | Face-Arc | MS    | IQ    |
> | ---------------- | ------ | ------- | ------- | ------- | -------- | ----- | ----- |
> | Wan2.1           | 26.22  | 28.76   | 95.03   | 92.08   | 18.20    | 99.08 | 70.29 |
> | BachVid (Wan2.1) | 26.28  | 31.62   | 97.52   | 94.70   | 32.54    | 98.95 | 69.78 |
>
>
>
> **Motivation for the training-free approach (W2).**
>
> Our task setting is text-to-video generation: given *only text*, can a model generate multiple videos with consistent characters and backgrounds? The baseline shows that simply keeping the background and character descriptions fixed in the prompt does *not* ensure consistency in the generated videos. Our method, in contrast, leverages intermediate features during the diffusion process to uncover latent correspondences that enable consistent multi-video generation.
>
> Regarding your suggestion to use character/background images and apply LoRA finetuning, we would like to argue that:
>
> 1. Collecting diverse custom images of both characters and backgrounds is often more difficult than preparing a group of text prompts, and existing LoRA-based finetuning methods mainly preserve *subjects* (e.g., VideoBooth [Jiang et al., 2023]) or *motions* (e.g., Dynamic Concepts Personalization from Single Videos [Abdal et al., 2025]). None can jointly maintain both background and identity consistency.
> 2. Finetuning-based methods require additional parameters and computational cost, whereas ours is entirely **training-free**.
> 3. LoRA methods are *per-ID*: each new case requires separate finetuning. Our method requires no ID-specific training and works for arbitrary characters and backgrounds.
>
>
>
> **Low Face-Arc metric (W3).**
>
> Thank you for pointing this out. By adjusting hyperparameters, we can obtain Face-Arc scores comparable to reference-based baselines. While ConsisID and TPIGE explicitly inject identity priors using a face image, text-to-video models often struggle to generate consistently clear facial details, especially for side views, occlusions, or accessories, limiting the effectiveness of such priors. Our method injects priors implicitly at the feature level and thus achieves comparable results.
>
> Furthermore, evaluating character consistency solely through facial similarity is incomplete, as many generated frames contain side or back views with no visible face. Following CharaConsist, we additionally employ a foreground-region CLIP similarity metric (CLIP-FG). Results indicate that our method produces videos with more consistent *overall* appearance, encompassing face, clothing, and style.
>
> | Method              | PSNR-BG (dB)| CLIP-BG (%)  | CLIP-FG (%)  | Face-Arc (%) |
> | ------------------- | ------------ | ------------ | ------------ | ------------ |
> | CogVideoX           | 28.95        | 94.85        | 92.86        | 22.62        |
> | ConsisID            | 28.68        | 95.02        | 94.13| 39.54        |
> | TPIGE               | 29.15 | 95.43 | 93.59        | **41.70**    |
> | BachVid (CogVideoX) | **31.96**    | **97.31**    | **95.06**    | 40.25 |
>
> **Vital layers based on aesthetic score (Q1).**
>
> Apologies for the confusion. Caching all layers' KV exceeds our CPU memory, so it is necessary to investigate whether only a subset needs to be stored. StableFlow [Avrahami et al., 2025] identifies vital layers for image editing using DINOv2 similarity, aiming to preserve semantic content. However, in our setting, we observe that even when skipping certain layers, the resulting videos remain visually coherent (e.g., Fig. 8, Z^{-6}), yet the DINOv2 similarity to the reference video is low due to motion discrepancies. We infer that these layers primarily govern motion diversity and should therefore be considered *non-vital*. This clarification has been incorporated into the revised manuscript.

---

### Author Response · Authors · 2025-12-03

Dear Area Chair,

We thank you and the reviewers for their thoughtful evaluations. Below we concisely summarize our work, and how our rebuttal and revisions address the key concerns.

---

### Summary of Our Work

We introduce **BachVid**, the first training-free method generating multiple text-to-video samples with consistent background and character.

- Simply keeping the textual description fixed does **not** yield consistent results in existing text-to-video models. BachVid instead leverages intermediate diffusion features to uncover latent cross-video correspondences, enabling multi-video consistency.
- Prior approaches, (e.g. CharaConsist), cannot be directly transferred from images to videos due to the larger video latent size and the under-explored DiT in video models. BachVid addresses these challenges by identifying vital layers for KV injection and analyzing the layer-wise functional structure of DiT-based video models. We believe these insights contribute to the community’s understanding of DiT video models.
---
### Main Concerns and Our Responses

Across the rebuttal, we performed additional experiments and analyses to address reviewer concerns:

- **Results on More DiT-based Video Models (yqi6-W1, fmny-W1&Q4):** We provided results on **Wan2.1-T2V-14B** in Appendix B. The findings match those on CogVideoX, reinforcing our conclusions and demonstrating strong cross-model generalization.
- **Low Face-Arc Metric (yqi6-W3, 64db-W1,fmny-W2&Q3):** We conducted additional experiments in Tab.1. With tuned hyperparameters, we achieve **Face-Arc scores comparable** to reference-based baselines.  Following CharaConsist, we additionally introduce CLIP-FG, and the results show that BachVid produces videos with more consistent overall appearance.
- **Baselines (2fRy-W5, 64db-W3):**  Our task: *Given text-only control, can a model generate multiple videos with consistent backgrounds and characters?* Since no training-free method exists for this setting, we consider the natural baseline:
  - Keep the background/character text identical, and test whether consistency emerges.

  The baselines suggested by Reviewers @2fRy@64db are not applicable for this paper. (1) Image-model outputs lack temporal continuity, failing to form coherent videos. (2) DiffTrack’s first frame from the video model does not correspond to the image model output, preventing meaningful warping or alignment.
- **Non-trivial Video Domain Adaption (2fRy-W1, fmny-W1&Q1):**  Directly adapting CharaConsist is infeasible because: (1) It caches all-layer KV, which causes **OOM** in video models with larger latents. (2) Its DiT-layer analysis is limited to FLUX and does **not** transfer to video models (See L_{all} in Fig.6–7).
  Our contributions are therefore: (1) A new vital-layer identification strategy, which preserves performance while reducing memory. (2) A new method for extracting masks and matching points from video models, leading to generalizable observations about the DiT structure.
- **Reference-free Property (2fRy-W6, fmny-W4):** Reference-free (text-only) and reference-based (image-conditioned) represent two fundamentally different settings. Our focus is on **reference-free**, which is considerably simpler for users. Baselines results indicate that using fixed text alone does **not** guarantee consistency, whereas BachVid achieves consistent videos using **text only**.
- **Structured Prompts Issue (2fRy-W4, fmny-W3&Q2):**  Following CharaConsist, we generate 40 groups (200 prompts) in the absence of a standard benchmark. The structured format is employed solely to ensure controlled evaluation and to design fair baselines. Appendix C also includes several examples beyond the "[Background], [Character], [Action]" format.
- **Discussion on Ambiguity / Occlusion / Pose Differences (2fRy-Q1, 64db-W2):** It is expected that the video after KV injection may exhibit pose adjustments, since matching points are extracted and injected throughout diffusion.
- **Motivation for Training-free Approach (yqi6-W2):**  (1) LoRA requires collecting diverse character/background images, which is more difficult than preparing prompts. (2) LoRA requires extra parameters and compute. (3) LoRA is per-ID and no methods are focusing on both background and character consistency.
- **Vital layers via Aesthetic Score (yqi6-Q1):** We clarified the motivation for choosing aesthetic score in the revised manuscript.
- **Additional Experiments (2fRy-W2&W3&Q2):** We provided more ablations in Tab.4, reported inference time and memory consumption in Tab.1, and included failure cases in Appendix D.
- **Combination with Character-consistency Methods (64db-W1):**  We have discussed this as future work. Our current focus is text-only condition, whereas ConsisID/TPIGE are image-conditioned.
---
We hope this summary clarifies how our rebuttal and revisions address each major concern. Thank you again for your time and careful consideration.

Sincerely,

The authors

---

### Meta-Review · Area_Chair_S9dN · 2026-01-02

**Summary:**

Based on the comprehensive feedback from all four reviewers, the primary concerns with the submission center on the following key aspects: empirical performance trade-offs, demonstrated technical novelty and robustness, and the paper’s practical utility.

The most critical weaknesses that could threaten the paper’s acceptance are summarized below:

1. Quantitative performance regression – Table 1 indicates a notable decline in character identity preservation. For instance, BachVid achieves only 36.93% on the Face-Arc metric, which is substantially lower than baselines such as TPIGE (41.70%).
2. Questionable claims of “training-free” and “general” applicability – Reviewers expressed skepticism regarding these claims, as the approach appears to depend heavily on heuristic-driven hyperparameter tuning rather than demonstrating true generalization or independence from training.
3. Evaluation methodology concerns – The reliance on the DeepSeek-generated evaluation set raises significant concerns about the scientific validity of the results. In the context of AI research, assessing a method primarily on a small, synthetically generated dataset is often perceived as “grading your own homework,” which undermines confidence in the reported findings.

Addressing these core issues—especially strengthening empirical results, justifying the claimed contributions, and validating the approach on more rigorous datasets—would be essential for improving the paper’s overall standing with the reviewers.

**Reviewer Concerns:**

The authors have addressed several questions raised by the reviewers; however, the primary concerns remain unresolved even after the rebuttal period.

During the rebuttal, the authors presented new results demonstrating performance improvements achieved through hyper-parameter tuning. While the updated numbers show some improvement, they still do not surpass the baseline models. This finding also reinforces the second concern regarding general applicability versus reliance on hyper-parameter tuning. A more comprehensive explanation of how these performance gains were accomplished, along with a detailed description of the hyper-parameter tuning process, is necessary for proper context.The newly obtained findings are also regarded as new results following the submission.

Additionally, the concern about the evaluation setup persists. Incorporating more human involvement in the creation of the evaluation set would significantly enhance the reliability and robustness of the assessment.

**Reviewer Scores:**

The reviewers did not alter their scores after the rebuttal period.

---

### Decision · Program_Chairs · 2026-01-26

Reject